# Respiratory modulations of cortical excitability and interictal spike timing in focal epilepsy: a case report

Daniel S. Kluger [1,2] ✉, Tim Erdbrügger[1], Christina Stier [1], Malte B. Höltershinken[1], Omid Abbasi[1], Martina Saltafossi[1], Kanjana Unnwongse[3], Tim Wehner[3], Jörg Wellmer[3], Joachim Gross[1,2] & Carsten H. Wolters [1,2]

## Abstract

**Background** Brain activity in focal epilepsy is marked by a pronounced excitation-inhibition (E:I) imbalance and interictal epileptiform discharges (IEDs) observed in periods between recurrent seizures. As a marker of E:I balance, aperiodic neural activity and its underlying 1/f characteristic reflect the dynamic interplay of excitatory and inhibitory currents. Recent studies have independently assessed 1/f changes both in epilepsy and in the context of body-brain interactions in neurotypical individuals where the respiratory rhythm has emerged as a potential modulator of excitability states in the brain.
**Methods** Here, we investigate respiration phase-locked modulations of E:I balance and their involvement in the timing of spike discharges in a case report of a 25 year-old focal epilepsy patient using magnetoencephalography (MEG).
**Results** We show that i) respiration differentially modulates E:I balance in focal epilepsy compared to $N = 40$ neurotypical controls and ii) IED timing depends on both excitability and respiratory states.
**Conclusions** These findings overall suggest an intricate interplay of respiration phase-locked changes in excitation and the consequential susceptibility for IED generation and we hope they will spark interest in subsequent work on body-brain coupling and E:I balance in epilepsy.

## Plain language summary

Epilepsy is a brain disorder in which abnormal electrical activity in the brain leads to seizures. We investigated the impact of breathing on electrical activity, particularly how breathing influences the balance between excitation and inhibition of electrical activity in the brain. We compared the impact of breathing patterns in a 25-year-old epilepsy patient on the excitation-inhibition balance with the effects seen in healthy individuals. We found there was a relationship between breathing and brain activity in the patient with epilepsy. We recommend further research be undertaken on how bodily rhythms impact epilepsy management, including the balance between excitation and inhibition in the brain.

Focal epilepsy is a neurological disorder marked by recurrent temporary seizure activity arising from a localised anatomic substrate, the so-called epileptogenic zone[1]. In addition, one specific alteration of neural activity in epilepsy patients is the generation of *interictal* spike activity, brief focal discharges occurring largely independently of seizures[2]. However, recent work has linked interictal epileptiform discharges (IEDs) and seizure activity more closely by revealing similar probability distributions of both signatures[3] and by demonstrating that IEDs are travelling waves arising from an epileptogenic source[4,5]. In general, one common observation in epilepsy concerns alterations in excitation-inhibition (*E:I*) balance: While constant E:I balance is considered essential for maintaining neural homeostasis[6] so that excitability within a particular neural array remains at a critical state[7], epilepsy is characterised by a critical imbalance introduced by

pathological upregulation of excitation. One neural signature that has emerged as a marker for E:I balance is aperiodic brain activity and its 1/f characteristic, meaning that low frequencies carry higher power. This dominant non-oscillatory component of brain activity reflects the dynamic interplay of excitatory and inhibitory currents[8] and its steepness (or slope) is an established read-out of cortical excitability[9]. Investigations of E:I balance have recently begun to apply 1/f measures in epilepsy[10,11] and are thus a useful extension of previous approaches demonstrating relative power shifts in high vs low oscillatory frequencies[12,13]. Collectively, these findings converge on the observation that E:I imbalance substantially contributes to the pathology of epilepsy[14–16].

In neurotypical cohorts, recent studies have demonstrated that both oscillatory[17,18] and non-oscillatory neural activity[19] are modulated by the

[1]Institute for Biomagnetism and Biosignal Analysis, University of Münster, Münster, Germany. [2]Otto Creutzfeldt Center for Cognitive and Behavioral Neuroscience, University of Münster, Münster, Germany. [3]Ruhr-Epileptology, Department of Neurology, University Hospital Knappschaftskrankenhaus, Bochum, Ruhr-University Bochum, Bochum, Germany. ✉e-mail: daniel.kluger@uni-muenster.de

breathing rhythm. One recurrent motif is the apparent systematic relationship between respiration and transient states of cortical excitability[20]. Extending previous findings of body-brain interactions in psychopathology (see[21] for review), there is an increasing number of accounts suggesting respiratory involvement in neuropsychiatric disorders[22,23]. By modulating disease-specific neural activity, respiration (and other physiological rhythms) may play a role in the course of certain disorders, particularly when they include alterations of excitability like schizophrenia or epilepsy.

Here, we aim to address a central gap in the previous literature by investigating respiration phase-locked modulations of E:I balance and their involvement in the timing of spike discharges in focal epilepsy. In light of previous evidence for i) respiration-related changes in E:I balance and ii) upregulation of excitability states in focal epilepsy, we hypothesised differential patterns of respiratory E:I modulation in a patient with focal epilepsy (compared to neurotypical controls). Furthermore, we expected the generation of IEDs to be systematically related to respiration-modulated changes in excitability within the epileptogenic zone. In keeping with these hypotheses, we report that respiration differentially modulates E:I balance in focal epilepsy compared to $N = 40$ neurotypical controls and that IED timing depends on both excitability and respiratory states.

## Methods

### Case description
A 25-year-old man sustained a perinatal left middle cerebral artery infarct, resulting in a left hemispheric porencephalic cyst and a spastic paresis of the right upper extremity. Seizures started at 8 years of age. The patient reported about one seizure with loss of awareness and oral automatisms per month. Previous video EEG recordings, however, suggested underreporting of seizures with subtle behavioural changes. The patient describes an involuntary gaze and head deviation to the right. Subsequent symptoms may include loss of awareness and tonic-clonic activity in all extremities. Scalp video EEG using 50 digitized electrodes according to the international 10-10 system revealed a monomorphic left frontocentral spike focus with electrographic maximum at electrodes FC1>Cz>C3 > CP1. Habitual seizures were recorded with onset in the left frontocentral region. Electrical source analysis (dipole and CLARA source models) based on 12 089 averaged spikes localised the centre of gravity to the left mesial frontal region, immediately adjacent to the wall of the porencephalic cyst. At the time of the recording, the patient was taking Brivaracetam 200 mg and Lacosamid 400 mg per day.

The patient and his family have agreed to publish the case description as stated above.

### Ethics statement
The study was conducted in accordance with the Declaration of Helsinki and approved by the institution's ethical review board (approval 25.05.2021, Ref. No. 2021-290-f-S). The patient gave written informed consent for this data to be used in scientific publications.

### Data acquisition and procedure
We simultaneously recorded MEG and respiratory data (6 runs of 8 min duration each) in a magnetically shielded room with the patient in supine position to reduce head movements. MEG data was acquired using a 275-channel whole-head system (OMEGA 275, VSM Medtech Ltd., Vancouver, Canada) at a sampling frequency of 2400 Hz. During recording, the patient was simply instructed to relax while lying down awake (see ref. 24). To minimise head movement, the patient's head was stabilised with cotton pads inside the MEG helmet. The patient was instructed to breathe naturally through the nose (and subsequently monitored via video) while the respiratory signal was measured as thoracic circumference by means of a respiration belt transducer (BIOPAC Systems, Goleta, USA) placed around his chest.

### Neurotypical control population
Forty healthy volunteers (21 female, age 25.1 ± 2.7 y [M ± SD]) were used as a neurotypical control population (data from[19]). All participants reported

having no respiratory or neurological diseases and gave written informed consent prior to all experimental procedures. The original study was approved by the local ethics committee of the University of Münster (Ref. No. 2018-068-f-S). Participants were seated upright in a magnetically shielded room while we simultaneously recorded 5 minutes of MEG and respiratory data with a sampling frequency of 600 Hz. Recordings took place in the same MEG scanner and with the same respiration belt as the patient recordings. Like the patient, control participants were instructed to breathe through the nose and monitored via video. During recording, participants were to keep their eyes on a fixation cross centred on a projector screen placed in front of them. The preprocessing and analysis steps described below were identical for patient and control data.

### MEG preprocessing
All MEG and respiratory data preprocessing was done in Fieldtrip for Matlab. Prior to statistical comparisons, we adapted the synthetic gradiometer order to the third order for better MEG noise balancing. Data were resampled to 300 Hz and power line artefacts were removed using a discrete Fourier transform (DFT) filter on the line frequency of 50 Hz and its harmonics (including spectrum interpolation). Finally, we applied independent component analysis (ICA) on the filtered data to capture and remove eye blinks and cardiac artefacts within the first 20 extracted components.

### Respiratory preprocessing
After resampling to 300 Hz, points of peak inspiration (peaks) and expiration (troughs) were identified in the normalised respiration time course using Matlab's *findpeaks* function. Phase angles were linearly interpolated from trough to peak ($-\pi$ to 0) and peak to trough (0 to $\pi$) to yield respiration cycles centred around peak inspiration (i.e., phase 0).

### Head movement correction
To rule out breathing-related head movement as a potential confound, we used a correction method initially established by Stolk and colleagues[25]. Based on online head movement tracking performed by our MEG scanner, six continuous signals represent x, y, and z coordinates of the head centre ($H_x$, $H_y$, $H_z$) as well as three rotation angles ($H_\psi$, $H_\theta$, $H_\varphi$) and hence together describe head movement. We constructed a regression model comprising the six 'raw' signals as well as their derivatives and, from these 12 signals, the first-, second-, and third-order non-linear regressors for a total of 36 head movement-related regression weights (using a third-order polynomial fit to remove slow drifts). This regression analysis was performed on power spectra of single-sensor time courses to remove signal components that can be explained by translation or rotation of the head with respect to the MEG sensors.

### Respiration-locked estimations of 1/f slope
We defined a parieto-occipital region of interest for movement-corrected MEG ($k = 41$ sensors; taken from ref. 19). In light of previous evidence for both respiration-locked changes of E:I balance in neurotypical individuals and the fundamental role of excitability in epilepsy symptomatology, we first investigated presumptive differences in the coupling of 1/f slope to the respiratory rhythm. This analysis was focussed on parieto-occipital cortex since i) it omits the patient's epileptogenic zone in order not to potentially conflate aperiodic and spike-related effects, ii) it has been identified as the site of strongest respiration-locked 1/f modulations in previous work[19], and iii) aperiodic neural activity is known to be governed by a posterior-to-anterior gradient[26,27]. Single-sensor time series of the entire recording within this ROI were entered into the SPRiNT algorithm[28] with default parameter settings and subsequently averaged. SPRiNT is an extension of the *specparam* algorithm[26] and uses a short-time Fourier transform (frequency range 1-40 Hz) to estimate the aperiodic component in neural time series within a moving window (1 s width, 75% overlap).

Thus, SPRiNT yielded time series of the aperiodic component (i.e., the 1/f exponent) with a temporal resolution of 250 ms and a frequency resolution of 1 Hz. In order to relate 1/f slope time series to the respiratory signal,

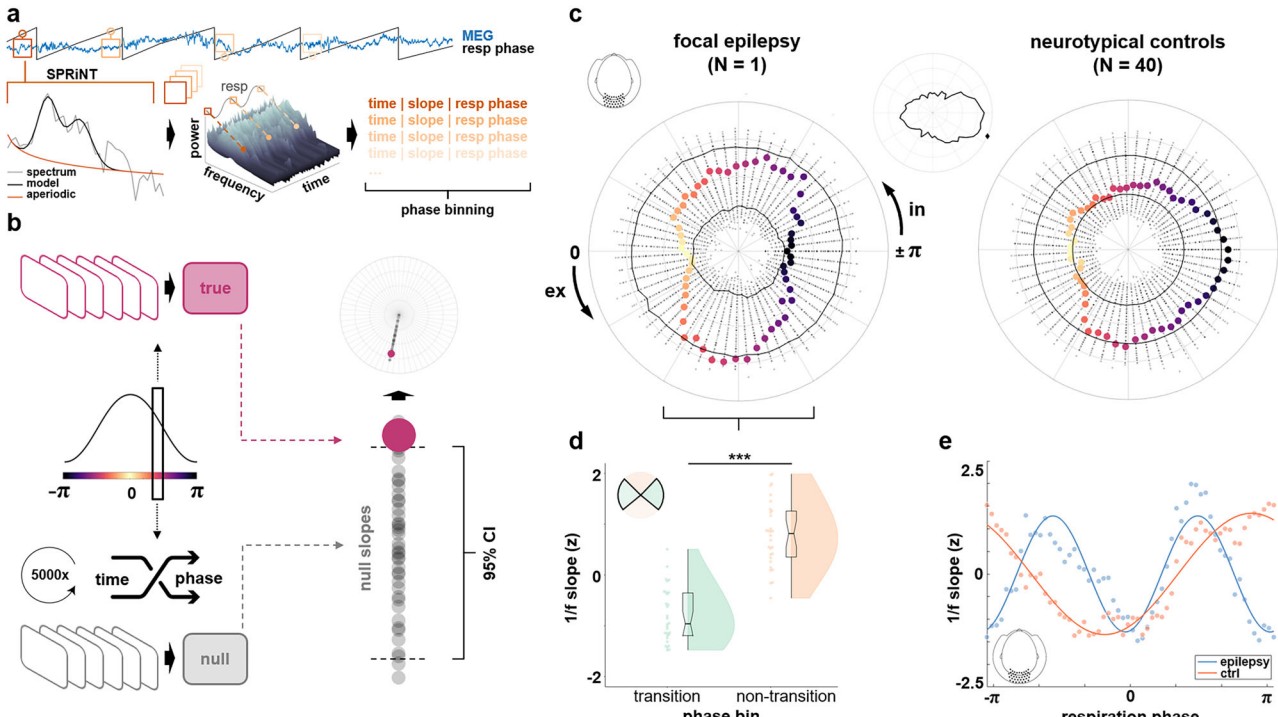

**Fig. 1 | Methods synopsis and differential modulations of posterior 1/f slope in focal epilepsy (vs neurotypical controls). a** Based on simultaneous recordings of whole-head MEG and respiration, we implemented the SPRiNT algorithm[28] and its moving window approach to estimate the aperiodic component in Fourier-transformed neural time series. Thus, for each time around which the moving window was centred, we could extract the corresponding 1/f slope estimate and momentary respiration phase. Phase binning then allowed us to compute bin-wise average 1/f slope to be used in statistical analyses. **b** For any given respiration phase, corresponding 1/f slopes were averaged to yield the empirical mean slope for that phase. Shuffling the data $k = 5000$ times to assign random respiration phases to the time points of SPRiNT computations (see (**a**)), the resulting phase averages were used to construct phase-specific null distributions of 1/f slope. Empirical slope estimates were then plotted in polar coordinates against their null distribution to illustrate 1/f slope over quasi-continuous respiration phase. **c** Polar visualisation of

1/f slope over parieto-occipital sensors across the respiration cycle in the focal epilepsy patient and $n = 40$ neurotypical controls. Coloured bold dots show respiration phase-dependent 1/f slope, radial scatter plots indicate null distributions of k = 5000 bin-wise group-level mean exponent values. Solid black lines indicate the 5th and 95th percentile of each bin's null distribution, respectively (see (**c**)). Small centre panel shows the difference of patient vs controls with the circular mean marked (~157 degrees). In = inspiration, ex = expiration. **d** In the epilepsy patient, 1/f slope was significantly flattened (indicating increased excitability) during the $n = 30$ inspiration-to-expiration and expiration-to-inspiration transitions (compared to $n = 30$ non-transition phase bins; Wilcoxon rank sum test: $z(29) = -6.22, p < 0.001$). **e** Ideal sine fits minimising the error sum of squares corroborated the differential modulation dynamics (see (**c**)) for the epilepsy patient (blue) compared to the group average of neurotypical controls (red).

we extracted respiratory phase at all time points for which slope had been fitted (i.e., the centres of each moving window). In keeping with previous work[20,29], the respiratory cycle was binned into $n = 60$ equidistant, over-lapping phase bins. Moving along the respiration cycle in increments of $\Delta\omega = \pi/30$, we collected 1/f slope fits computed at a respiration angle of $\omega \pm \pi/10$. On these estimates, we then computed bin-wise averages of 1/f slope, yielding quasi-continuous 'phase courses' of the aperiodic component. In order to compute confidence intervals for the empirical slope estimates, we first constructed a surrogate respiration time series based on the iterated amplitude-adjusted Fourier transform (IAAFT[30]). In contrast to shuffling the respiration time series, this iterative procedure preserves the temporal autocorrelation of the signal. From these IAAFT-transformed respiration time series, we then extracted surrogate respiratory phase values corresponding to each 1/f slope estimation from SPRiNT. As with the empirical data, we finally binned all SPRiNT outputs into n = 60 (surrogate) phase bins and computed bin-wise average slope exponents over surrogate respiration phase. This procedure was repeated 5000 times and resulted in a null distribution of 5000 surrogate 1/f slope estimates × 60 phase bins.

### Statistics and reproducibility

*Analysis of 1/f slope.* We used a Hodges-Ajne test provided by the circstat toolbox[31] to evaluate whether the difference vector of patient vs control courses of 1/f slope would be uniform across the respiration cycle. Separately for patient and control data, we then used a Wilcoxon rank sum test to

compare ROI-averaged 1/f slopes during respiratory transition phases (i.e., inspiration to expiration and vice versa) and non-transition phases (i.e., during ongoing inspiration and expiration, respectively). Transition phases (see Fig. 1d) were defined as $[-\pi/4$ to $\pi/4]$ (inspiration-to-expiration transition) and $[\frac{3}{4}\pi$ to $-\frac{3}{4}\pi]$ (expiration-to-inspiration). Aiming to quantify the difference in distributions of 1/f over respiration phase between patients and controls, we used Matlab's *fit* function to obtain ideal linear sine fits for both slope distributions.

**IED-locked statistical analysis.** Time points of IED peaks were marked in the raw MEG time series by an experienced epileptologist (KU). For the IED-locked analysis of 1/f slope and respiration phase, we first identified a set of $n = 6$ sensors which most clearly showed spike activity in the raw data. In addition to visual inspection of the raw data, we quantified the amplitude of spikes as the variance within each sensor's normalised time series in a 250 ms time window around the spike peak. We computed the cumulative density function across all sensors (sorted by accumulated variance across spikes) and defined a cutoff at the elbow of that distribution. For the resulting region of interest, we computed event-locked 1/f slope estimates from single-sensor time-series ± 1000 ms around the spike peaks using the approach outlined above. Averaging across individual sensors yielded ROI-average slope estimates for 1000 ms-time windows centred around lags of −500 ms, −250 ms, 0 ms, 250 ms, and 500 ms relative to the peaks of $n = 74$ spikes.

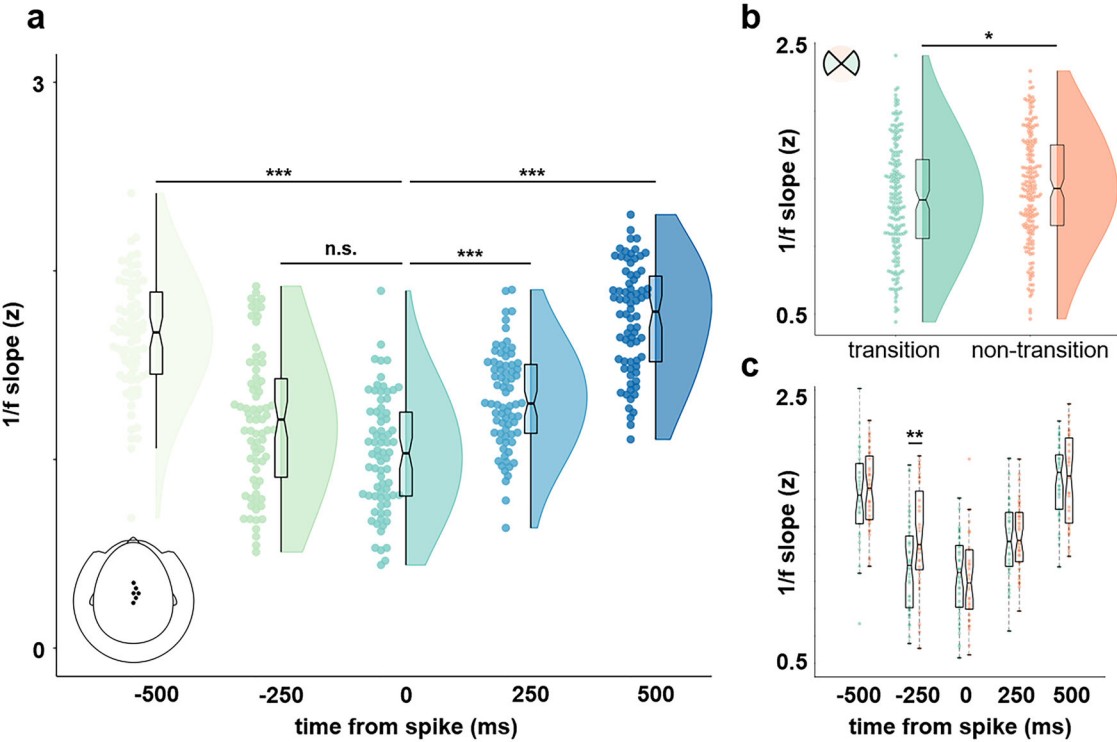

**Fig. 2 | E:I balance and respiration modulate spike timing. a** Over sensors which most clearly showed interictal spikes in the raw data, 1/f slope was significantly flattened prior to the generation of $n = 74$ spikes. **b** The LMEM revealed a small but significant decrease in slope for time points around spike peak ($\pm 500$ ms) occurring at a respiratory transition phase (vs non-transition). **c** Excitability was particularly increased for transition-phase events prior to spike peak (at a lag of $-250$ ms).

Mean slope differences across lags were assessed with a Friedman test and subsequent single comparisons (using Matlab's *multcompare* function). To analyse the effects of lag and respiration phase on 1/f slope, we set up a model comparison of two linear mixed-effects models (LMEMs). The first (base) model only included the lag effect and was defined (in Wilkinson notation) as

$$\text{slope} \sim \text{lag}^2 - \text{lag} + (1|\text{spike}) \qquad (1)$$

For each spike event, the model predicted 1/f slope estimates as a combination of intercept and the quadratic fixed effect of lag. The random effect for spike was included to allow for variation in modulation effects across single spikes. For comparison, a second LMEM including categorical respiratory information (transition vs non-transition bin) for each spike and lag plus select interaction terms was defined as

$$\text{slope} \sim \text{lag}^2 - \text{lag} + \text{phase bin} + \text{lag}^2{:}\text{phase bin} + (1|\text{spike}) \qquad (2)$$

Thus, the second model yielded coefficients for the quadratic fixed effect of lag and fixed effect of respiration phase bin as well as interaction effects. We used Matlab's *compare* function to test whether including respiratory information would significantly increase the LMEM fit (accounting for the additional effects) by means of a theoretical likelihood ratio test.

### Reporting summary
Further information on research design is available in the Nature Portfolio Reporting Summary linked to this article.

## Results
### Respiration differentially modulates E:I balance in focal epilepsy
We computed 1/f slope in moving windows (see ref. 28 across the full length of continuous MEG recordings (Fig. 1a). Relating the slope estimate in each window to the corresponding respiratory phase allowed us to identify systematic changes of slope across the respiratory cycle. While 1/f slope significantly modulated by respiration in both data sets (Hodges-Ajne tests: both $p < 0.001$), we found a distinct difference in the modulation shape of the epilepsy patient compared to neurotypical controls: Around the expiration-to-inspiration transition, the group of neurotypical controls showed a steeper slope, i.e. a decrease in parieto-occipital excitability during late expiration. In contrast, the epilepsy patient showed an excitability *increase* during late expiration, observed as a flatter slope around the expiration-to-inspiration transition (Fig. 1c). A Hodges-Ajne test confirmed that the difference between both slope distributions was not uniform ($p < 0.001$), but focussed towards the end of the breathing cycle (circular mean = 157 degrees, Fig. 1c). Overall, 1/f slope was significantly flattened during both transition phases in the patient's data (Wilcoxon rank sum test: $z = -6.22, p < 0.001$; see Fig. 1d) but not in neurotypical controls ($z = 0.86$, $p = 0.39$). Consequently, the patient's resulting shape of 1/f slope over respiration exhibited a "double frequency" characteristic (with flat slope around both phase 0 and $\pm\pi$) compared to the controls (flat slope around phase 0 and steep slope around $\pm\pi$). This impression was corroborated by fitting sinusoidal models to the respective data sets (Fig. 1e) which showed high goodness of fit for patient (adjusted $R^2 = 836$) and control data (adjusted $R^2 = 908$).

### Spike timing depends on excitability and respiratory states
In order to integrate neural and physiological modulations of epileptic brain signatures, we investigated the generation of interictal spikes (marked by an experienced epileptologist) as a function of both excitability states and respiration phase. Using the continuous MEG recordings, we determined a cluster of $n = 6$ right-lateralised fronto-central electrodes in which the interictal spikes were most clearly visible (Fig. 2a). For these sensors, we then computed ROI-average time-resolved fits of 1/f slopes in an interval of $\pm 500$ ms around the peak amplitude of $n = 74$ spike discharges. We observed a strong U-shaped relationship

between 1/f slope and the lag relative to spike peak (Fig. 2a), in that 1/f slope significantly increased at longer lags both before and after spike peak (Friedman test for the $\pm$ 500 ms interval: $\chi^2(4) = 154.45$, $p < 0.001$). At spike peak (lag = 0), excitability was significantly increased compared to all positive and negative lags (single comparisons: all $p < 0.001$) except for time points preceding IED generation (i.e., lag = −250 ms: $p = 0.133$), suggesting that IEDs occurred at time points of particularly high excitability (i.e., flattened 1/f slope).

Since we had previously found excitability to be modulated by respiration, we hypothesised that the generation of interictal spikes would similarly depend on respiratory phase. In a final analysis, we thus extracted respiration phase time-locked to spike onset as well as the negative and positive lags. Using a linear mixed effects model (LMEM) to predict 1/f slope as a function of lag relative to spike centres, we first verified the quadratic effect shown in Fig. 2a ($t(368) = 18.24$, $p < 0.001$). To account for the potential influence of respiration, we repeated the LMEM to not only include the lag effect but also information about respiration phase (*transition* vs *non-transition* bins, see Fig. 1d). This second model yielded significant effects of lag ($t(365) = 11.74$, $p < 0.001$) and respiration phase bin ($t(365) = -2.01$, $p = 0.045$), indicating that both neural (excitability) and physiological states (respiration) modulate the generation of IEDs. Model comparison of the two LMEMs confirmed that including respiratory information significantly improved the model fit ($\chi^2(3) = 12.93$, $p = 0.005$). While we found an overall flatter slope for time points corresponding to respiratory transition (vs non-transition) phases ($z = 1.92$, $p = 0.028$, one-tailed Wilcoxon rank sum test, Fig. 2b), this overall effect was clearly driven by the time interval preceding spike onset (Welch's $t(72) = 2.42$, $p = 0.009$, Bonferroni-corrected $p = 0.045$; see Fig. 2c). Fittingly, the LMEM yielded a significant interaction of respiration phase bin (transition vs. non-transition) and the linear effect of lag (i.e., from pre- to post-spike time points; $t(365) = 2.89$, $p = 0.004$). Hence, the shift of E:I balance towards hyperexcitability was most pronounced when the pre-spike interval coincided with the transition of respiration phase.

## Discussion

In this case report, we investigated respiration phase-locked modulations of E:I balance and their involvement in the timing of interictal epileptiform discharges (IEDs) in focal epilepsy. Van Heumen and colleagues[10] previously demonstrated measures of aperiodic brain activity as an indicator of neuropathophysiology in an MEG case study of childhood focal epilepsy. In the present case study, we validate aperiodic 1/f slope as a useful marker of E:I balance in focal epilepsy and extend these findings by highlighting two key aspects of altered body-brain coupling: First, characteristic hyperexcitability could be observed in the patient's coupling between respiration and non-oscillatory read-outs of excitation-inhibition balance. Compared to neurotypical controls, breathing-related increases of cortical excitability in this particular patient were observed twice as frequently, namely during *both* inspiration-to-expiration and expiration-to-inspiration transitions. Second, not only did interictal epileptiform activity occur during periods of particularly high excitation, but pre-spike E:I upregulation was specifically linked to respiratory transition phases. This triadic link between breathing, excitability, and spike activity, including the close connection between ictal and interictal activity[4,5], connects to the long-standing observation that seizures in childhood generalised epilepsy can reliably be triggered by hyperventilation[32–34]. Despite the clear need for dedicated and fully powered studies, one central mechanism hypothesised to provoke these seizures is respiratory alkalosis, i.e. breathing-related transient states of elevated arterial pH[35]. Fluctuations in $CO_2$ are inversely related to tissue acidity (pH) with lowered pH causing an increase in extracellular adenosine[36]. Following the proposal that $CO_2$-induced changes in neural excitability are caused by pH-dependent modulation of adenosine and ATP levels[37], Salvati and colleagues[38] recently demonstrated pH sensitivity in seizure-generating circuits of a seizure-prone rodent model for absence epilepsy. Critically, much of the presently available evidence comes from long-term respiratory interventions, whereas investigations of cycle-by-cycle variation in $CO_2$ remain scarce. While we and others have previously discussed such pH-related changes in a non-clinical context as one potential driver of respiration-related modulations in neural activity[29,39], future work should specifically target the question of short-term $CO_2$ variation and its potential role in modulating neural activity. At this point, mechanistic considerations are critically informed by the distinction of nasal vs oral breathing pathways: Respiratory coupling of neural oscillations is dependent on initial oscillatory activity in the olfactory bulb (see[22]), as evident from the absence of coupling in bulbectomised animals[40]. Accordingly, respiratory modulations of brain oscillations dissipate during oral breathing in humans[41]. For non-oscillatory activity, on the other hand, we recently found no differences in respiration phase-locked effects between nasal and oral breathing[19]. We, therefore, suggested that respiratory links to oscillatory vs. non-oscillatory brain activity may reflect different underlying mechanisms of central-peripheral coupling. In line with rodent work by Salvati and colleagues[38], the present case study in focal epilepsy suggests an intricate interplay of respiration phase-locked changes in excitation and the consequential susceptibility for IED generation.

## Conclusions

While this case report is certainly limited in its generalisability, we hope it will spark interest in subsequent work on body-brain coupling and E:I balance in epilepsy. Here, it is important to acknowledge the complexity and diversity within the group of epilepsy disorders. Conceivably, it will be instrumental to consider how both type-specific alterations (e.g. network connectivities with strong links to the epileptogenic zone) as well common denominators across epilepsy types (e.g. E:I imbalance and IEDs) systematically interact with peripheral rhythms.

## Data availability

Fully anonymized data that support the findings of this study are available from the first author pending a formal data sharing agreement. The source data for Fig. 1 and Fig. 2 can be found in Supplementary Data 1 and 2, respectively.

## Code availability

The analysis code is publicly available through the Open Science Framework (OSF) via https://osf.io/z4xgp/?view_only=54c85247e7164a22b554175e873edcc5.

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

## Acknowledgements
The authors would like to thank Karin Wilken, Ute Trompeter, and Hildegard Deitermann for their assistance during data collection. D.S.K. is supported by the IMF (KL 1 2 22 01) and the DFG (KL 3580/1-1). J.G. is supported by the DFG (GR 2024/11-1, GR 2024/12-1). T.E., M.B.H., and C.H.W. are supported by the Bundesministerium für Gesundheit (B.M.G.) as project ZMI1-2521FSB006, under the frame of ERA PerMed as project ERAPERMED2020-227, and by the DFG (WO 1425/10-1). We acknowledge support from the Open Access Publication Fund of the University of Münster.

## Author contributions
Conceptualisation, D.S.K.; Methodology, D.S.K., J.G.; Investigation, T.E., T.W.; Data Curation, T.E., K.U.; Formal Analysis, D.S.K.; Writing—Original Draft, D.S.K., C.S., K.U., T.W., J.G., C.H.W.; Writing—Review & Editing, D.S.K., T.E., C.S., M.B.H., O.A., M.S., K.U., T.W., J.W., J.G., C.H.W.; Visualisation, D.S.K., T.E., C.H.W.; Funding Acquisition, D.S.K., J.G., C.H.W.

## Funding

## Competing interests
The authors declare no competing interests.
