## [Transparent Peer Review file · Communications Medicine]

Respiratory modulations of cortical excitability and interictal spike timing in focal epilepsy - a case report

Corresponding Author: Dr Daniel Kluger

Version 0:

Reviewer comments:

Reviewer #1

(Remarks to the Author)

The case report by Kluger and coworkers describes MEG data from a patient with pharmacoresistant epilepsy following a perinatal vascular insult. The authors use the 1/f paradigm, an indirect proxy for E/I-balance, to assess excitability and report that the modulation of E/I-balance by respiration is different from that in healthy controls (n=40). Moreover, interictal discharges are entrained by respiration as well as by E/I-state.

This is an interesting report providing a window into network dysregulation in chronic epilepsy as well as an example of body-to-brain signaling with some translational potential (relations between respiration and epilepsy are an important, potentially under-studied field of pathophysiology). The manuscript is well written, controls with 40 individuals provide a very good background, and findings are clearly documented.

My minor remarks are:

(1) It would be good to show the 1/f plots in Fig. 1a a bit larger, maybe even in a comparison between patient and controls. If possible, having the respiratory signal in the 3D-diagram would be great.

(2) Did the authors check for respiration through the mouth? Several aspects of respiratory brain modulation depend on breathing through the nose, and it would be extremely interesting to know whether this is the case for the 1/f phenomenon or for IEDs as well.

(3) A large part of the discussion focusses on breathing-related changes in pH, which may account for changes in excitability. However, much of the quoted work refers to longer phases of hypo- or hyperventilation, a well-known paradigm in, e.g., febrile seizure. Whether or not a cycle-to-cycle variation in pH is causally related to respiratory EEG/MEG changes should be related more specifically (following the authors' own work, as quoted).

Reviewer #2

(Remarks to the Author)

The authors have investigated the influence of the respiratory cycle on the excitation-inhibition balance and the timing of interictal discharges in a focal epilepsy patient using magnetoencephalography, comparing the results with those from forty neurotypical controls. Although the paper is well written, it is severely limited by its nature as a single case report. Indeed, it seems unjustified to me how the authors can expect to perform statistical analysis comparing a single patient to 40 healthy controls. Important clinical characteristics of the patient are not described (e.g., current anti-seizure medications, seizure frequency, seizure type). All of these clinical variables are extremely important, especially antiseizure medications, since they are known to modulate the electrical activity of the brain (Höller, Yvonne, Christoph Helmstaedter, and Klaus Lehnertz. "Quantitative pharmaco-electroencephalography in antiepileptic drug research." *CNS drugs* 32.9 (2018): 839-848.). Moreover, the novelty of this study is not clearly presented, as the influence of respiration (in particular hyperventilation) on the modulation of interictal epileptiform discharges is already well-known (Yamatani, Miwa, et al. "Hyperventilation activation on EEG recording in children with epilepsy." *Pediatric neurology* 13.1 (1995): 42-45, among many others).

There are several other concerns (listed below, not in order of importance):

Introduction: The authors do not state a clear scientific hypothesis or explain why they used magnetoencephalography to achieve their scientific aim. Instead, they prematurely provide the results of the present study.

Methods: It is unclear whether the patient or healthy subjects were recorded during sleep or awake states. It is also unclear

which segment was analyzed for the patient (I assume a specific analysis was performed for IEDs and then for background activity, which, in fact, is for some reasons explained in the Results section). However, it is not stated how much of the background activity was analyzed (from Figure 1, it seems the whole recording was considered). Have the authors performed some sort of preprocessing/filtering?

Results: The first paragraph of the results is just a specification of methods, therefore it should be moved to the previous section.

Results: As previously stated, my main issue with this work is that the proposed results based on a single patient are not generalizable to the entire population of people with epilepsy. Therefore, any potential clinical or neurophysiological implications based on comparing the patient with the healthy controls are completely speculative. This is the main non-correctable issue that I have with this work.

Discussion: The discussion is generally well-written, but as previously stated, all the conclusions reported are speculative and not supported by the present results. Several sentences are not relevant to the case reported; for example, the authors discuss in detail the literature on childhood epilepsy, but the case they analyzed involves a 25-year-old adult.

The authors are encouraged to perform a systematic evaluation of the proposed methodology in a specific group of people with epilepsy compared to the already analyzed healthy controls, with a clear scientific hypothesis, to actually provide evidence of the presented findings.

Version 1:

Reviewer comments:

Reviewer #1

(Remarks to the Author)

The authors have adequately and convincingly addressed my concerns. I have no further comments or critique.

I now went through the comments of reviewer 2 and the authors's answers and changes in the ms.

The reviewer's main points are i) that a case study (which it basically is, given that only one patient has been included) cannot be generalized and ii) that changes in ventilation and their effects on EEG in epileptic patients have already been extensively studied, reducing the novelty of the work.

I fully and decidedly agree with the author's responses to this critique. They overtly agree (and now make even more clear) that the work is a case study. It is up to the journal to decide whether such studies should be published, but there is certainly nothing wrong with this established design. The second issue is based on a misunderstanding: the authors studied the cycle-to-cycle coupling of MEG to respiration, a phenomenon which has gained much attention during the past ten years and has not yet been analyzed in epilepsy. The study thus opens a new and promising window into pathophysiology and, potentially, diagnosis/prediction, and it clearly has an added value for the community.

All other points are either reiterations of the previous two criticisms or more technical, and have been adequately addressed.

Therefore, I think that both reviews have been well addressed, including the points by reviewer 2.

Response to reviewers

We thank the reviewers for their insightful comments and provide a point-by-point response to their suggestions below. We would also like to explicitly express our appreciation for the editorial decision to value a single-case report as a proof-of-principle for novel analyses. In what follows, the reviewers' comments are presented in grey, our responses in blue, changes made during the revision in red.

Reviewer #1:

The case report by Kluger and coworkers describes MEG data from a patient with pharmaco-resistant epilepsy following a perinatal vascular insult. The authors use the $1/f$ paradigm, an indirect proxy for E/I-balance, to assess excitability and report that the modulation of E/I-balance by respiration is different from that in healthy controls ($n=40$). Moreover, interictal discharges are entrained by respiration as well as by E/I-state.

This is an interesting report providing a window into network dysregulation in chronic epilepsy as well as an example of body-to-brain signaling with some translational potential (relations between respiration and epilepsy are an important, potentially under-studied field of pathophysiology). The manuscript is well written, controls with 40 individuals provide a very good background, and findings are clearly documented.

We appreciate the reviewer's very positive evaluation of our work.

My minor remarks are:

(1) It would be good to show the $1/f$ plots in Fig. 1a a bit larger, maybe even in a comparison between patient and controls. If possible, having the respiratory signal in the 3D-diagram would be great.

Thank you, we appreciate this suggestion. The $1/f$ plot has been increased in size and a respiration trace has been added to the 3D plot to further underscore the logic of the analysis. Please note that the aperiodic fit shown in Fig 1a merely serves illustrative purposes and does not actually show results of any particular analysis:

(2) Did the authors check for respiration through the mouth? Several aspects of respiratory brain modulation depend on breathing through the nose, and it would be extremely interesting to know whether this is the case for the 1/f phenomenon or for IEDs as well.

The reviewer is entirely correct in pointing out the distinct effects of nasal vs oral breathing, many of which we have discussed in detail in previous work (Kluger et al., Nat Comm 2023). This is why it was important for us to rule out oral breathing in the present study which we accomplished by video monitoring both the patient as well as the control participants. A systematic comparison of (non-)oscillatory changes following nasal vs oral breathing has yet to be conducted in epilepsy patients and is one of the key factors in a larger study currently in preparation.

In order to avoid any ambiguities, the Methods section now explicitly reports video monitoring:

The patient was instructed to breathe naturally through the nose (and subsequently monitored via video) while the respiratory signal was measured as thoracic circumference by means of a respiration belt transducer (BIOPAC Systems, Goleta, USA) placed around his chest. (p. 3)

Like the patient, control participants were instructed to breathe through the nose and monitored via video. (p. 4)

(3) A large part of the discussion focusses on breathing-related changes in pH, which may account for changes in excitability. However, much of the quoted work refers to longer phases of hypo- or hyperventilation, a well-known paradigm in, e.g., febrile seizure. Whether or not a cycle-to-cycle variation in pH is causally related to respiratory EEG/MEG changes should be related more specifically (following the authors' own work, as quoted).

We thank the reviewer and have revised the Discussion section to make this critical distinction more transparent:

Following the proposal that CO₂-induced changes in neural excitability are caused by pH-dependent modulation of adenosine and ATP levels (Dulla et al., 2005), Salvati and colleagues (2022) recently demonstrated pH sensitivity in seizure-generating circuits of a seizure-prone rodent model for absence epilepsy. **Critically, much of the presently available evidence comes from long-term respiratory interventions, whereas investigations of cycle-by-cycle variation in CO₂ remain scarce. While we and others have previously discussed such pH-related changes in a non-clinical context as one potential driver of respiration-related modulations in neural activity (Kluger & Gross, 2020; Xu et al., 2011), future work should specifically target the question of short-term CO₂ variation and its potential role in modulating neural activity.** (pp. 10-11)

Reviewer #2:

The authors have investigated the influence of the respiratory cycle on the excitation-inhibition balance and the timing of interictal discharges in a focal epilepsy patient using magnetoencephalography, comparing the results with those from forty neurotypical controls. Although the paper is well written, it is severely limited by its nature as a single case report. Indeed, it seems unjustified to me how the authors can expect to perform statistical analysis comparing a single patient to 40 healthy controls.

We want to thank the reviewer for their overall positive evaluation of our work. We fully acknowledge the limits of case reports regarding their generalisability and discuss this aspect in greater detail in response to the reviewer's comments below.

Important clinical characteristics of the patient are not described (e.g., current anti-seizure medications, seizure frequency, seizure type). All of these clinical variables are extremely important, especially antiseizure medications, since they are known to modulate the electrical activity of the brain (Höller, Yvonne, Christoph Helmstaedter, and Klaus Lehnertz. "Quantitative pharmaco- electroencephalography in antiepileptic drug research." *CNS drugs* 32.9 (2018): 839-848.). Moreover, the novelty of this study is not clearly presented, as the influence of respiration (in particular hyperventilation) on the modulation of interictal epileptiform discharges is already well-known (Yamatani, Miwa, et al. "Hyperventilation activation on EEG recording in children with epilepsy." *Pediatric neurology* 13.1 (1995): 42-45, among many others).

We thank the reviewer for pointing out two important issues, namely clinical details of the patient (especially medication) and hyperventilation. We have now added the suggested details to the 'Case description' section of the manuscript, including a description of the seizures and medication at the time of the recording.

As for the medication issue, we would like to note that baseline changes in brain activity due to medication cannot explain the *systematically different coupling to respiration*. The temporal dynamics of respiratory coupling to E:I balance we investigate are not affected by potential alterations in neural activity alone.

The revised paragraph now reads as follows:

The patient reported about one seizure with loss of awareness and oral automatisms per month. Previous video EEG recordings, however, suggested underreporting of seizures with subtle behavioural changes. (p. 3)

At the time of the recording, the patient was taking Brivaracetam 200mg and Lacosamid 400mg per day. (p. 3)

Regarding respiratory modulation of IEDs, we agree with the reviewer that the novelty of our study could have been phrased more clearly. As Reviewer #1 notes in their final comment (see above), respiratory influence on neural changes in epilepsy has so far been exclusively investigated in the context of *long-term changes of respiratory patterns* (e.g. hyperventilation), be they voluntary (provocative) or pathological (symptomatic). It is entirely unknown, however, to what extent cycle-by-cycle variation of neural activity during *natural breathing* - as quantified by measures of excitation-inhibition balance, for example - is altered in patients with focal epilepsy.

In response to the comments made by both reviewers, the Discussion section has been rephrased accordingly (kindly see our response to Reviewer #1 above):

Following the proposal that CO₂-induced changes in neural excitability are caused by pH-dependent modulation of adenosine and ATP levels (Dulla et al., 2005), Salvati and colleagues (2022) recently demonstrated pH sensitivity in seizure-generating circuits of a seizure-prone rodent model for absence epilepsy. **Critically, much of the presently available evidence comes from long-term respiratory interventions, whereas investigations of cycle-by-cycle variation in CO₂ remain scarce. While we and others have previously discussed such pH-related changes in a non-clinical context as one potential driver of respiration-related modulations in neural activity (Kluger & Gross, 2020; Xu et al., 2011), future work should specifically target the question of short-term CO₂ variation and its potential role in modulating neural activity.** (pp. 10-11)

In addition, the Yamatani reference has been added to the manuscript.

There are several other concerns (listed below, not in order of importance):

Introduction: The authors do not state a clear scientific hypothesis or explain why they used magnetoencephalography to achieve their scientific aim. Instead, they prematurely provide the results of the present study.

We thank the reviewer and have revised the Introduction section in order to state our hypotheses more clearly. The brief summary of key results has been removed as suggested. The corresponding paragraph now reads as follows:

Here, we aim to address a central gap in the previous literature by investigating respiration phase-locked modulations of E:I balance and their involvement in the timing of spike discharges in focal epilepsy. **In light of previous evidence for i) respiration-related changes in E:I balance and ii) upregulation of excitability states in focal epilepsy, we hypothesised differential patterns of respiratory E:I modulation in a patient with focal**

epilepsy (compared to neurotypical controls). Furthermore, we expected the generation of IEDs to be systematically related to respiration-modulated changes in excitability within the epileptogenic zone. (pp. 2-3)

Methods: It is unclear whether the patient or healthy subjects were recorded during sleep or awake states. It is also unclear which segment was analyzed for the patient (I assume a specific analysis was performed for IEDs and then for background activity, which, in fact, is for some reasons explained in the Results section). However, it is not stated how much of the background activity was analyzed (from Figure 1, it seems the whole recording was considered). Have the authors performed some sort of preprocessing/filtering?

Thank you for pointing out these ambiguities - all participants were recorded in an awake resting state and the computation of 1/f slope was indeed performed on the entire recording to yield a time-resolved measure of E:I balance. All preprocessing is described in section 2.5 ('MEG preprocessing') and did not include filtering except for a 50Hz DFT filter to remove line noise.

In response to the reviewer's comment, the Methods section has been revised as follows:

During recording, the patient was simply instructed to relax while lying down **awake** (see Baud et al., 2018). (p. 3)

Single-sensor time series **of the entire recording** within this ROI were entered into the SPRiNT algorithm (Wilson et al., 2022) with default parameter settings and subsequently averaged. (p. 5)

We computed 1/f slope in moving windows (see Wilson et al., 2022) across the **full length of continuous MEG recordings** (Fig 1a). (p. 7)

Results: The first paragraph of the results is just a specification of methods, therefore it should be moved to the previous section.

The section has been moved as suggested.

Results: As previously stated, my main issue with this work is that the proposed results based on a single patient are not generalizable to the entire population of people with epilepsy. Therefore, any potential clinical or neurophysiological implications based on comparing the patient with the healthy controls are completely speculative. This is the main non-correctable issue that I have with this work.

We certainly agree with the reviewer that results from a case report cannot be generalised to the population level, which is why we clearly stated our hope that this proof-of-concept study would inspire follow-up research to test specific hypotheses. Given the novelty of i) respiratory modulation of E:I in general, ii) cycle-by-cycle respiration-brain coupling in focal epilepsy, and iii) the link between IED generation and natural respiration *in the absence of respiratory interventions* like hyperventilation, we strongly feel that a well-described case report has great merit in providing first evidence for potentially very fruitful work on specific aspects.

We would argue that this is particularly true when novel methodology (here: time-resolved estimates of E:I balance) is first introduced to a new field. Our manuscript was among the very

first clinical applications of the novel SPRiNT algorithm to estimate dynamic changes in excitability states, and as such constitutes a valuable proof of principle for subsequent research.

As an additional comment, as we note in the manuscript, generalisation across the entire population of epilepsy patients will be difficult to attain due to the complex variety of subtypes within the family of epilepsy diagnoses.

Discussion: The discussion is generally well-written, but as previously stated, all the conclusions reported are speculative and not supported by the present results. Several sentences are not relevant to the case reported; for example, the authors discuss in detail the literature on childhood epilepsy, but the case they analyzed involves a 25-year-old adult.

Despite the merit of case studies we have outlined in response to the reviewer's previous comment, the generalisability of their results is of course limited. We have revised the Discussion section in order to clearly delineate where further research is needed. Although our conclusions are phrased with caution, they are supported by the present results. The reason we discuss childhood epilepsy is the immediate link of that literature to CO₂-related mechanisms of E:I changes through respiratory modulation. The scarcity of previous reports in adults underscores the importance of further research - as we state in the Discussion, we hope that the proof-of-concept provided by this case study will inspire future applications of respiratory E:I modulation in patients with epilepsy. Finally, in response to the reviewer's earlier comment regarding long- vs short-term respiratory effects, we hope to have made a clearer distinction between the cycle-by-cycle influence of natural breathing and long-term artificial alterations of respiratory behaviour like hyperventilation.

In response to the reviewer's comment (and complementing our response to suggestions made above), the Discussion section has been revised as follows:

In the present case study, we validate aperiodic 1/f slope as a useful marker of E:I balance in focal epilepsy and extend these findings by highlighting two key aspects of altered body-brain coupling: First, characteristic hyperexcitability could be observed in the **patient's** coupling between respiration and non-oscillatory read-outs of excitation-inhibition balance. Compared to neurotypical controls, breathing-related increases of cortical excitability **in this particular patient** were observed twice as frequently, namely during *both* inspiration-to-expiration and expiration-to-inspiration transitions. (p. 10)

Despite the clear need for dedicated and fully powered studies, one central mechanism hypothesised to provoke these seizures is respiratory alkalosis, i.e. breathing-related transient states of elevated arterial pH (Laffey & Kavanagh, 2002). (p. 10)

In line with rodent work by Salvati and colleagues (2022), the present **case study** in focal epilepsy suggests an intricate interplay of respiration phase-locked changes in excitation and the consequential susceptibility for IED generation. (p. 11)

The authors are encouraged to perform a systematic evaluation of the proposed methodology in a specific group of people with epilepsy compared to the already analyzed healthy controls, with a clear scientific hypothesis, to actually provide evidence of the presented findings.

As this closely relates to the reviewer's previous comment above, we would like to briefly reiterate that we fully acknowledge the need for fully powered group studies and we are currently preparing them. However, the limited interpretability of case studies does not negate their benefits as proof-of-concept or proof-of-principle research, particularly when i) prior evidence is limited, ii) the methodology is new and has to be introduced to the field in a comprehensible application, and iii) the pattern of findings clearly suggests that follow-up studies will likely further our understanding of the pathology.